# Barriers of West African women scientists in their research and academic careers: A qualitative research

Ndèye Marème Sougou[1,2]*, Oumy Ndiaye[3], Farah Nabil[4], Morenike Oluwatoyin Folayan[4,5], Samba Cor Sarr[6], Elhadji M. Mbaye[7], Guillermo Z. Martínez-Pérez[4]*

1 Department of Preventive Medicine and Public Health, University Cheikh Anta Diop, Dakar, Senegal, 2 Institute of Health and Development, University Cheikh Anta Diop, Dakar, Senegal, 3 Faculty of Economics and Management, University Cheikh Anta Diop, Dakar, Senegal, 4 Faculty of Health Sciences, University of Zaragoza, Zaragoza, Spain, 5 Obafemi Awolowo University, Ile-Ife, Nigeria, 6 Ministry of Health and Social Action, Dakar, Sénégal, 7 Institut de Recherche en Santé, de Surveillance Épidémiologique et de Formation, Diamniadio, Sénégal

* ndeyemareme.sougou@ucad.edu.sn (NMS); gmartinezgabas@hotmail.com (GZMP)

**Data Availability Statement:** Data cannot be shared publicly as per the authors agreement with the National Ethics Committee of Senegal. The transcripts and audio files containing the study

## Abstract

### Objective

This study aims to identify barriers to the professional advancement of women researchers in West Africa.

### Methods

This was a descriptive, observational, cross-sectional qualitative study conducted between June and September 2020 in five West African countries (Ghana, Senegal, Burkina Faso, Niger and Mali). Interviews were conducted with 21 female and 9 male health researchers by video call. After transcription, the data was thematically analysed using an inductive process.

### Results

Four themes associated with barriers to women's careers development were identified. First. was family- and environmental-related barriers. Gender norms that assign domestic tasks and responsibilities to women reduced the time they were able to dedicate to research. Second was gender insensitive organisational culture and institutional policies that deepened gender disparities and made it more difficult for women to attain leadership positions. Third was the need for women in research to undergo emancipation programs to strengthen their resilience and ability to make critical decisions as strategic approaches to address the challenges faced by women in the academia were a lot more focused on addressing their relationship with their spouse. Forth, was the individual intermediate perception of professional and personal success which for many women, they perceive themselves as competent as their male counterparts and should not be subject to the gender discrimination they experience.

data are confidential and cannot be shared unless the interested party is a member of the research team or signs a justified confidentiality agreement. For data access, please contact: Permanent Secretary of the National Health Research Ethics Committee Senegal. Phone: +221 773614212. Email: cnrs2008@live.fr.

**Funding:** This study is a compenent of the BCA-WA-ETHICS project, part of the EDCTP2 programme, supported by the European Union under grant number: CSA2018ERC- 2314. The grant was awarded to authors GMZP, SCS, and EMM. The funders had no role in study design, data collection and analysis, decision topublish, or preparation of the manuscript. Full name of the funder: The European and Developing Clinical Trials Partnership (EDCTP), https://www.edctp.org/.

**Competing interests:** The authors have declared that no competing interests exist.

## Conclusion

The web created between work-life and home-life for West African women researchers mainly as a result of the gender inequalities in the social structure will require more medium- and long-term strategic planning by institutional authorities to reduce gender disparities in research and academia. This work has highlighted the influence of family and social life on the professional lives of West African women researchers. The study could help contribute to the development of gender equality interventions for the career development of women researchers in West Africa.

## Introduction

The participation of women in research is increasing as evidenced by data emerging from the global north. Statistics from Elsevier indicate that the proportion of female scientists in the European Union is 41% while women's representation among investors is only 12% [1]. Despite the progress made, women and girls continue to be under-represented in the fields of science, technology, engineering, and mathematics (STEM) [2]. Also, women researchers are still increasingly under-represented as they move up the ladder of an academic career [1]. The proportion of women represented in academia also varies depending on the domain, with more women in the life and health sciences while men were more dominant in the physical sciences. Also, women tend to spend more time on teaching and less time on research; in addition, women usually apply for smaller grants than men [3]. Furthermore, fewer women occupy leadership positions such as research coordinators or principal investigators [3].

Identified barriers to women's participation in research include the absence of gender equity and gender supportive strategies in many research and academic institutions [3, 4]. Other barriers include gender bias in the employment process which implicitly favours men, poor institutional monitoring of women's representation in research and academia, and the lack of institutional strategies to support work-life balance for women researchers [1]. Furthermore, women reportedly faced greater delays in completing their doctoral studies and producing a strong publication record, which could be attributed to gender and social norms [4].

In Africa, statistics on the percentage of women researchers show alarming disparities. In 2016, the overall percentage was 34%, but this figure hides great disparities that exist on the continent according to the African Development Bank [5]. In Cape Verde, for instance, 52% of researchers are women, 47% in Tunisia, and 40% in both of South Africa and Uganda. Guinea is at the other end of the spectrum with only 6% of researchers being women, followed by Ethiopia, where the rate is 7.6%, then by Mali with 10.6%, and Côte d'Ivoire with 16.5% [5]. The same dynamic is observed in Senegal, where women are poorly represented in the research community and even more so in the decision-making bodies of academic and research institutions [6]. In 2019, women represented only 29.3% of the academic staff, countrywide and all disciplines combined [7]. In Niger, women represented 10% of university teaching staff in 2005 [8], in Ghana, they represented 20% of health researchers in 2010, in Burkina Faso they represented 27.7% of personnel in medical science research in 2010, and in Mali women constituted only 14.9% of health researchers in 2006 [8].

Some efforts had been invested to improve women's representations in institutions in Africa by the development of gender friendly policies. These include the of the Gender and Affirmative Action Implementation Centre which was mandated to implement gender policies in some institutions such as Kenyatta University [9]; the effort at improving women's access to

research funding through the establishment of the "Projet d'Appui à la Promotion des Enseignantes Chercheures du Sénégal" (PAPES) [10] and the establishment of an International Center for Girls' and Women's education in Africa by the African Union and Microsoft located in the French-speaking African countries [11] Despite these efforts, gender main-streaming in development policies and programs are still insufficient, and Francophone West Africa remains one of the regions with the greatest gender disparities in the education sector [11]. Women still face major obstacles in accessing higher education [12].

The limiting sociocultural expectations placed on women's role in research and academia can strongly affect their performance, and in turn, their opportunities for professional development [13]. Yet, women's participation in global research is essential. Women's participation in research increases the prospect of women's specific health issues being addressed [7]. In addition, there are important moral, ethical, and justice arguments in favour of diversity in research as diverse teams are reported to generate more innovative solutions to problems and provide a more holistic view in general [13, 14]. There has however been limited discussion about the gendered processes in organisations that have led to the productions of the phenomenon described so far thereby leading to an assumption that research institutions and academia are gender-neutral and asexual in its practices [15]. The poor recognition of the way sexuality had shaped the work processes in academic institutions and the imprint of cultural practices on these processes have also been poorly described, discussed and addressed. This has largely arisen from the grounding of organisational processes in the *working worlds and relations of men* [16]; the failure for organisational structures and structural reforms to be shaped by feminist perspectives [17]; and the failure to acknowledge that disrespectful gender behaviours emanate from the enshrined gender culture of the institution are not just the result of an individual anomaly but that gender blind institutional policies, procedures and interaction birth these practices [15].

This study not only acknowledges the growing body of literature on women in research but a dearth in the academia especially in Africa. There is a growing recognition of the masculine genderization of the operations of the research institutions [18]; and there may be an assumption of such genderized operations of African research organisations because of the patriarchal societies found in Africa. There has however been some progress towards gender equality howbeit this progress is not uniform across the globe especially in the Science, Technology, Engineering, and Maths (STEM) faculties; and the differences in expressions by context is also less often discussed. These differences in expression need to be understood to be able to develop context specific responses to the inequalities experienced in different contexts and settings.

This study built on Fagenson's theory that acknowledges that women's career advancement in any organization is influenced by societal and systematic factors; and that cultural and social attitudes toward gender responsibilities influences perceptions of job roles and responsibilities [19]. We also recognize that women's representation in science and technology is often systematically minimized, undercut, and undercounted through a genderization process that impacts their employment opportunities; the narrative of their contributions [20]; and impacts access to family-related support or predisposes them to family-related discrimination [21]. Many African communities the family and societal environment promote gender subordination [22] and thus, where socio-cultural norms operates within an institutional framework often place women under male tutelage [23]. Gender bias and stereotype tendencies persist within these African institutions [24].

The question is whether research and academic institutions in West Africa reproduce societal gender relations and representations and whether they influence gender constructs at the organizational and individual levels with an impact on the career advancement of women

researchers. In order to fill this gap in knowledge, this study aims to identify barriers to the career advancement of women researchers in some West African countries. The study aims to identify ways to improve on the inequitable representation of women in scientific research, therefore, contributing to higher diversity among researchers, which will, in turn, lead to higher quality research.

## Materials and methods

### Study design

This was a descriptive, observational, cross-sectional qualitative study. Data collection was done between June and September 2020.

### Study population

The study population consisted of female and male scientists working in academic and research institutions that promote, design, conduct and/or disseminate biomedical, clinical and socio-epidemiological research in West Africa. Study participants were included if they reside in West Africa, can read and communicate in French, Portuguese or English (the three main official languages in the West African region), are members of academic or research institutions in West Africa working in health or social sciences discipline, and are over 18 years of age.

### Sampling

Participants were recruited by non-probability sampling. The sample size was determined as the data was collected and analysed in an iterative process [25]. The aim was to seek a richness of information. Thus, the size was only known when saturation occurred and no more new information had been generated by interviews.

### Data collection and analysis

In-depth individual interviews were conducted by video call using the Zoom platform. Consent forms were sent to the informants 24 hours in advance to obtain their signatures on the consent form. All interviews were recorded. An interview guide with open-ended questions was used during all interviews. Each interview lasted for an average of 60 minutes. The discussion guide explored six main topics as indicated in Table 1: information about the disease, relationship with healthcare professionals, daily life, social support, mood, and the future.

The data analysis was analysed by creating a theoretical corpus from the full transcripts of the interviews. The interviews were transcribed as the data was collected from the digital recording. The authors used thematic analysis through an objective and systematic analysis of the manifested content within the oral discourse [26]. An inductive analysis was done to explore the research objects [27]. After the transcription, the data was organised into evocative themes in relation to the interview.

The analysis of the coded data followed the following steps: 1) extraction of the different units of analysis from the participants' discourse using the Nvivo software, 2) horizontal analysis of the theoretical corpus in relation to the research objectives, 3) creation of the theory from the analysis of the coded data, and 4) validation of the meaning of the statements by triangulation of sources and methods. The data analysis was carried out using the NVivo 12 software.

**Table 1. Interviews guide.**

| | **Please, we would like to listen to your opinions and experiences on:** | |
|---|---|---|
| 1 | Professional career of researchers | *Can you tell me about your professional career as a researcher?* |
| 2 | Opinion on gender equality in the field of research | *What do you think about gender equality in the field of research?* |
| | | *How do you judge the policies of African academies in terms of gender equality?* |
| | | *What are the organizations in African academies and research institutes doing in terms of gender equality?* |
| 3 | Opportunities related to the gender and/or sex of the researcher | *What opportunities for development have you had during your career as a researcher.* |
| | | *(Probe discussions on opportunities received related to the researcher's gender)* |
| | | *What is the comparison you can make between the trajectory of a female researcher and that of a male researcher (in terms of opportunities)?* |
| 4 | Barriers related to the gender and/or sex of the researcher | *What barriers have you encountered during your career as a researcher? -* |
| | | *(Explore any theme on discrimination due to the researcher's gender in the academic environment)* |
| | | *What are the environmental barriers that may impede the careers of women researchers? (Explore about the rôle of family and the social environment)* |
| 5 | Difficulties of women researchers in their professional careers | *What are the difficulties encountered by women researchers in attaining positions of responsibility?* |
| | | *Can you share any personal experiences? (Ask question if discussant identified difficulties with career development)* |
| | | *How would you compare the trajectory of a female researcher with that of a male researcher (in terms of difficulties)?* |
| 6 | Recommendations for achieving gender equality in universities and research institutes | *Can you give us recommendations for improving gender equality in universities and research institutes?* |
| | | *What are the coping strategies employed by women researchers in the face of the difficulties and obstacles they encounter in advancing their careers?* |
| | | *Are there other coping strategies that women researchers can adopt to build their careers in research?* |

## Ethics

Participation in this study was voluntary. Informed consent was obtained from all participants. The consent form was prepared and shared prior to any interview in the language chosen by the participant. The data collected was kept confidential. The identity of individuals who consented to participate was not included in the transcripts. The audio recordings were anonymised. In all uses of the results, anonymity was respected, with no identifying information on the notes and analysis materials. No remuneration was provided to the informants. The approval of the Senegalese National Health Research Ethics Committee was obtained for this study (0000050/MSAS/DPRS/CNERS).

## Results

Interviews were conducted with 21 female researchers and 9 male researchers from Ghana, Senegal, Burkina Faso, Niger and Mali (Table 2). The age of the participants ranged from 30 to 56-years-old, their years of experience as researchers ranged from 5 years to over 30 years, and they all had a PhD or its equivalent.

**Table 2. Sex-disaggregated and sociodemographic characteristics of participants.**

| Participants | Female N = 21 | Male N = 9 |
|---|---|---|
| Marital status | | |
| Single | 1 | 0 |
| Married | 19 | 9 |
| divorced | 1 | 0 |
| Study level | | |
| PhD | 19 | 9 |
| PhD Student | 2 | 0 |
| Master | 0 | 0 |
| Religion | | |
| Muslim | 18 | 8 |
| Christian | 3 | 0 |
| Atheist | 0 | 1 |
| Age | | |
| 25–35 years | 2 | 2 |
| 35–45 years | 14 | 3 |
| >45 years old | 5 | 4 |
| Years of professional experience | | |
| 0–5 years | 3 | 2 |
| 5–10 years | 11 | 5 |
| Over than 10 years | 7 | 2 |
| Country Representation | | |
| Senegal | 18 | 7 |
| Ghana | 1 | 0 |
| Bénin | 1 | 2 |
| Burkina Faso | 1 | 0 |

Though participants had diverse professional and personal life experiences, the four themes that resonated throughout the interviews reflected their perception of gender inequality in academia and research institutions irrespective of their country of origin as highlighted in Table 3.

## Organisational culture and institutional policies and practices

Organisational culture and institutional policies and practices affect opportunities for women to advance their career. Organisational culture is defined as the underlying values, beliefs and principles that serve as the foundation of an organisation's management system, as well as the set of management practices and behaviours that both exemplify and reinforce these core principles [28]. Most of the interviewees identified that few women are working in most research institutions and organizations in West Africa. This discrepancy in gender representation in research institutions reflects the under-representation of women in schools in general and less so a situation that arises from discriminatory recruitment. Indeed, few women reached this level of university education where they could compete with men.

*"At the University of Ziguinchor, at least for the example of 162 people, we are only 10 women professor-researchers".*

(Female, 40 years old, Professor-researcher)

Institutional discriminatory practices were discreet and implicit. A number of the interviewees were not consciously aware of discriminatory practices in the organizational systems

**Table 3. Summary of the thematic analysis of the main theme definitions and sub-theme categories.**

| Themes | Sub-topics |
|---|---|
| **Organisational culture and institutional policies** | Barriers: gender unequal environment, office politics, organisational barriers, workload, institutional policies, faculty track, promotion limitations, attrition |
| | Facilitators: leadership development, progress on diversity and inclusion, progressive change, opportunities for advancement, continuous training, opportunities for women |
| **Family and environmental barriers** | Barriers: position of women in the family, the position of women in society, unequal work-life balance, subordination in marital relationships |
| | Facilitators: prioritisation of work and private life, relationship, equitable roles and responsibilities in the marital relationship |
| **Individual characteristic—intermediate perception of professional and personal success** | Barriers: lack of confidence, personal limitations, behaviours based on perceived expectations, challenging gender biases, gender as a barrier |
| | Facilitators: self-efficacy, self-advocacy, hard work, career satisfaction, professional growth, positive reputation, gender as an opportunity |
| **Resilience strategies of women researchers in the face of obstacles to career advancement** | Perception of difficulties, career stagnation, slow career progress, personal strategies for dealing with obstacles, systemic strategies for dealing with obstacles |

and so discounted these practices as norms, and therefore were less able to report them during the interviews. Identified institutional gender discriminatory practices include the relegation of some subordinate roles to women such as taking meeting minutes while visible leadership roles were reserved for men.

Also, interviewees reported that the marital status of the female researcher plays a determining role in her being given leadership roles. For example, many leadership roles cannot be held by unmarried–single, divorcee–women; as being unmarried reportedly implied the woman has not acquired enough managerial skills to run an office. Also, unmarried women who hold leadership positions were touted as having gotten the job through sexual favouritism.

*"I was a general secretary in a university union. So, I coordinated a trade union section and I got a number of negative things from that. For example, people said, 'Because she doesn't have a husband, she has time to fight for everything and anything. I also had to fight to safeguard my reputation which at one point was tainted by nasty rumours about how I got into that job...",*

(Female, 49 years old, Professor-researcher)

Some discrimination was also reported concerning the right to maternity while men are not denied the right to paternity in any situation or circumstances.

*"I was in a project in London, the project lasted two years and the motto was: no baby for two years. You're not allowed to have a child, you're not allowed to..."*

(Female, 50 years old, Professor-researcher)

Many interviewees also reported difficulties in finding opportunities to advance their careers. They report that there are few gender-focused grants. In addition, they, like men, are confronted with the lack of funding opportunities for research in Africa.

## Family and environmental barriers

This theme deals with the social environment that affects the freedom of creative expression of individuals in society. A society shares a set of morals and traditions and is characterized by collective activities, interests and behaviour [29]. When a member deviates from the established norms and patterns of the group, his or her behaviour is considered subversive and threatens the stability and security that others derive from group membership. This theme describes how the family and social environment interferes with the career development of women researchers.

Interviewees identified the gender roles ascribed to women in the family and society as a barrier to research career advancement. Women are nurtured to be home carers and thus, they play a central role in the life of their families as a wife and/or mother. The responsibility to care and cater for the needs of the immediate and extended family–the children, husband, and elderly–is time-consuming and leaves little time to address research career advancement needs including missing multiple opportunities for travels, collaborations, networking, and capacity building.

Husbands played a decisive role in the future of the wife's professional career. Many interviewees identified with the need to have an understanding husband to be able to have a successful research career. A researcher narrated how she was forced to give up a travel grant (a traineeship in a research department in Europe) after a year in order to 'look after her husband' when she was not yet a mother. Her husband had demanded she return. Another researcher noted:

*"I had two maternities. It was very, very difficult. Then we have mobility grants that we sometimes can't use because we don't want to leave the country. We have children,. . .it was really an obstacle for me."*

(Female, 37 years old, Professor-researcher)

The majority of the interviewees emphasised the difficulty of achieving a work-life balance as their life balance was skewed towards prioritizing raising and caring for the family due to the socio-cultural values placed on women as homemakers in West African. Some female researchers, therefore, chose to put their research careers on hold while supporting the growth and development of the family.

Male researchers corroborated this narrative of women prioritizing family responsibilities. This family support enabled men to devote quality time to their professional work. They recognized that the prioritization of family responsibilities–a responsibility not equitably shared with men—by women was at the expense of their research career.

*"If the little family is happy, God is happy"*

(Female, 42 years old, Professor-researcher)

*". . .to women researchers, we say, "Find enough time. It's not your husband's job to look after the children or the kitchen. So they feel guilty. In fact, we need to change our mentality as African women because I don't think that's how it is in Europe. . .".*

(Female, 42 years old, Professor-researcher)

The in-laws were also reported to influence the researcher's career. The in-laws were there to remind the woman researcher of her social duty towards her family and in-laws.

*"Sometimes when my husband agrees that I can travel for scientific conferences, it is the in-laws who say that it is not possible, you can't be away all the time, etc."*

(Female, 45 years old, Professor-researcher)

Interviewees also identified that women also have to handle societal roles assigned to women, from which female researchers were not exempt. Women were expected to attend family ceremonies and socialize with the extended family. These social rules resulting from the gender roles ascribed to women in the African society further depletes the time women can devote to research.

"*We have social obligations although being a female academic, you have to go to christenings, to that kind of stuff. . .*"

(Female, 48 years old, Professor-researcher)

## Intermediate perception of professional and personal success

This theme deals with individual characteristics, which describe the dispositional, habitual and motivational traits discernible in an individual and which can help explain and predict certain behaviours [30]. Individuals vary in how they perceive and respond to obstacles. The women interviewed felt as competent as the men and felt that competency should be the main measure for assessment in the research enterprise.

*"But sometimes it turns out that the woman has as many skills as the men, um, but nevertheless she is rejected because she is a woman. So if she has the same skills as the men, that's it and everything, and she has the knowledge, she must be given the place she deserves.*

(Female, 36 years old, Professor-researcher)

Sadly, this is not the case. Many participants indicated that women have to cross multiple hurdles during their research career that have to do with perceptions of womanhood. This made what is an already inherently difficult field more challenging to manage as a woman. Women had to do a lot more to prove themselves and are consistently required to show commitment and professionalism because their work patterns do not conform to the academic research culture prescribed by men. This forced many women to conform to certain practices at the cost of personal values and interests. Women also needed to negotiate the gender stereotypical labelling of their moods and characters been modulated by female hormones as the quote below indicates:

*"I know that anyway, the female touch can be really good. . . But, whenever there are a lot of women in a place sometimes. . . They are often a bit 'knife-edged, I don't know if it's hormones or what [laughs].*

(Male, 40 years old, researcher)

*"A woman should not speak loudly, should not challenge. . .It is expected that this is exactly the same in the professional spheres and this is problematic.*

(Female, 36, Professor-researcher)

### Resilience strategies of women researchers

The notion of 'sacrifice' came up several times in the discourse of women researchers. For most women, sacrifices had to be made in order to succeed in an academic career.

"The academic career requires a lot of sacrifices because what you have to do at university you have to do. You have to do it because you chose it. So nobody else is going to do it for you."

(Female, 40 years old, Professor-researcher)

In the bid to fulfil both workplace and home-related responsibilities, some women researchers indicated that they preferred to work at night after finishing their household chores. Others, however, acknowledged that they had to work at their own pace leading often to a delay in the progression with their research career when compared to their male peers. Marriage was therefore considered a major obstacle to research career development by the interviewees.

Interviewees identified strategies to address the limitations that the social institution of marriage placed on their research career pathways. Some women researchers were happy to be divorced in order to pursue their careers with little stress.

*"Marriage, in fact, is an institution that blocks women's matrimonial careers, uh because I have found that the women who are the most successful are often women who have been married and who are finally relieved of the burdens of marriage, relieved of the problems of the husband or the in-laws. . ."*

(Female, 36 years old, Professor-researcher)

Other women researchers expressed they have chosen to be in polygamous households, as being the second wife would allow them the time to address their research career needs when their husbands are with the other wife or wives. For some, it was a question of entering a form of social homogamy with their spouse wherein they identified spouses with whom they shared the same professional ideologies. This would guarantee them spousal support for their career development as they would understand the time investment needed. For a smaller number of women informants, the solution was to marry later in life, after they had achieved a certain status (advancement/position of responsibility) in their research career.

## Discussion

In Africa, research on gender relations is deeply rooted in the mechanisms underlying the exclusion, domination or marginalization of women in society and/or in the workplace [31, 32]. This research is part of this framework and explores the barriers in the academic and research environment that impede women's full participation and development. Though we observed that women in West Africa also had slower research career development like their counterparts in the Global North, the reasons for the slowness may differ to an extent.

Family and environmental barriers remain the greatest barriers to women's participation and progression in research. This result confirms those of Milewski et al. who consider that women often struggle to find a balance between work and family life, which is a factor of precariousness for these women [33]. Our study has shown that women researchers find it difficult to find time to devote to research and to progress in the field. They are often subject to family responsibilities that require them to confine themselves to domestic activities and

devote less time to research-related activities. This corroborates with Sayer who consider that women spend far more time on unpaid domestic work than men [34]. In addition, the subordination of women in African households [35] implies that women have to seek their husbands' approval and support to make a success of their research career. These constraints may have led several potential female researchers, who are less able to adopt the enumerated coping strategies identified by interviewees, to stay away from the research enterprise. This reaction may be a reason for the persistent gender disparities in research as a career in West Africa, especially in the field of biomedical research [31]. This merits further research and explorations.

Women also struggle to navigate their careers [36]. The study also illustrated the effect of organizational influences such as male-dominated networks, bullying and harassment [36] as research institutions in West Africa reproduce and mirror the acts of gender inequalities in society [37]. These institutions have organisational and institutional cultures that meet the needs of male professors and students [38]. Leadership patterns, beliefs, symbols, structures, ceremonies, power and information flows are modelled on male expectations and experiences [38]. Women, therefore, had the additional task of negotiating the male-dominated practices in the research enterprise as much as they had to negotiate for home-front support from spouse and family. For many women, these negotiations are too hard a price to pay and, therefore, are forced to relegate the career tussle to conform to gender norms and values. A few academic institutions, however, have programs supporting gender equity. Existing programs primarily target the individual or interpersonal level of the social ecological interaction [39].

The discourse on strategies to address challenges women face with their career trajectory development centred on spousal related actions that indicate the need for the empowerment of women to be able to emancipate themselves. Female emancipation refers to distancing oneself from certain family, matrimonial, or statutory constraints reserved for women, while empowerment refers to the multidimensional socio-political process of women themselves, who are individually and collectively aware of the relations of domination that they are seeking to transform [40]. Some women entrepreneurs in Togo, who, in order to devote themselves to their careers, have expressed the wish that their husbands take a second wife [41] as a polygamous union enabled them to relegate domestic tasks to a younger woman, thereby allowing them the needed time to devote to their careers [41]. Divorce was less often mentioned as a solution or a way of reclaiming their personal trajectory [42]. This may be due also to the concerns about societal stigma associated with not being married. In addition, not being married may compromise the career development they require, as *unmarriedness* deters from them being entrusted with leadership roles. This circle of events indicates the complexity of the decisions a female researcher in West Africa may have to make for the purpose of career advancement. While empowerment programs may help strengthen women's resolve to overcome the barriers to their research progress in the short term, more medium and long-term strategies are needed to overcome the limitations placed by the gender inequalities rooted within the research and academic structures.

One of the strengths of the study was the conduct of the research exploration across West Africa, a region with the same cultural values and perspectives. The findings, therefore, become applicable to the region and can facilitate support by regional bodies. The study, however, has a few limitations. The data collected was limited to self-reported information on research and academic advancement and gender-related experiences. The coding and analysis of this data were conducted in this context. Additional themes and perspectives derived from the content-rich narratives of the participants merit further exploration. The experiences of empowerment of women researchers in their homes would also benefit from further exploration in future studies. The data also covers only 5 of the 15 countries in West Africa. Despite

these limitations, the results provide findings of a regional context that informs on the gender disparity observed in the career trajectory of women in research, thereby, adding context to the global discussion on promoting gender equity in global research career development.

## Conclusion

A complex set of factors has contributed to the gender inequality experienced by women in universities and research institutions in West Africa. The same complexity is reflected in the decision-making process by women researchers in an effort to extricate themselves out of the limiting web and make a success of their career. The gender insensitive organizational culture and institutional policies of these universities and research institutions make the extricating process laborious for women thereby causing many to succumb to the pressure to conform to gender expectations which jeopardizes their career development. A gender-sensitive change in organizational culture and institutional policies and practices may likely result in a significant increase in career research progression for women in research in West Africa. It is the authors' hope that this research will contribute to the development of a theory on the barriers and facilitators to the academic and professional advancement of researchers as well as to the development of evidence-based interventions to close the gender-related gaps in West African academic and research institutions.

## Acknowledgments

The authors are grateful to the research participants for their collaboration to achieve the research objectives.

## Author Contributions

**Conceptualization:** Ndèye Marème Sougou, Guillermo Z. Martínez-Pérez.

**Data curation:** Ndèye Marème Sougou.

**Formal analysis:** Ndèye Marème Sougou.

**Funding acquisition:** Samba Cor Sarr, Guillermo Z. Martínez-Pérez.

**Investigation:** Ndèye Marème Sougou, Oumy Ndiaye.

**Methodology:** Ndèye Marème Sougou, Samba Cor Sarr, Guillermo Z. Martínez-Pérez.

**Project administration:** Ndèye Marème Sougou, Farah Nabil.

**Resources:** Samba Cor Sarr, Elhadji M. Mbaye.

**Software:** Ndèye Marème Sougou, Oumy Ndiaye.

**Supervision:** Farah Nabil, Elhadji M. Mbaye.

**Validation:** Ndèye Marème Sougou.

**Visualization:** Ndèye Marème Sougou.

**Writing – original draft:** Ndèye Marème Sougou, Oumy Ndiaye.

**Writing – review & editing:** Ndèye Marème Sougou, Farah Nabil, Morenike Oluwatoyin Folayan, Samba Cor Sarr, Elhadji M. Mbaye, Guillermo Z. Martínez-Pérez.

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
