## [Decision Letter · Decision Letter 0]

21 Jul 2021

PONE-D-21-20760

Barriers of West African women scientists in their research and academic careers: A qualitative research

PLOS ONE

Dear Dr. Sougou,

Thank you for submitting your manuscript to PLOS ONE. After careful consideration, we feel that it has merit but does not fully meet PLOS ONE’s publication criteria as it currently stands. Therefore, we invite you to submit a revised version of the manuscript that addresses the points raised during the review process.

We look forward to receiving your revised manuscript.

Kind regards,

Teresa Carvalho

Academic Editor

PLOS ONE

Journal Requirements:

2. During our internal checks, the in-house editorial staff noted that you conducted research or obtained samples in another country. Please check the relevant national regulations and laws applying to foreign researchers and state whether you obtained the required permits and approvals. Please address this in your ethics statement in both the manuscript and submission information.

3. When reporting the results of qualitative research, we suggest consulting the COREQ guidelines: http://intqhc.oxfordjournals.org/content/19/6/349. In this case, please consider including more information on the number of interviewers, their training and characteristics; and please provide the interview guide used.

Additional Editor Comments:

This is a very timely and relevant topic. The paper is relevant and of interest to the journal readers. However, it still needs some minor revisions in order to be accepted for publication. Authors should take into detail consideration the recommendations made by the two reviewers. It is particularly relevant that the authors redesign the theoretical framework.

Reviewers' comments:

Reviewer's Responses to Questions

**Comments to the Author**

1. Is the manuscript technically sound, and do the data support the conclusions?

Reviewer #1: Yes

Reviewer #2: Yes

2. Has the statistical analysis been performed appropriately and rigorously? 

Reviewer #1: N/A

Reviewer #2: N/A

3. Have the authors made all data underlying the findings in their manuscript fully available?

Reviewer #1: Yes

Reviewer #2: Yes

4. Is the manuscript presented in an intelligible fashion and written in standard English?

Reviewer #1: Yes

Reviewer #2: Yes

5. Review Comments to the Author

Reviewer #1: This paper addresses a very interesting, under researched topic and specially targeting this geographical context. I believe the authors have very relevant data which can enhance knowledge on the topic addressed but it deserves further (theoretical) discussion in order to better contribute to this domain of knowledge and be published. Therefore, I suggest to improve the theoretical review at the beginning – introduction. The topic on the mechanisms underlying the exclusion, domination or marginalization of women in society and/or in the workplace, preventing them to fully progress in their career is very important and this work contributes a relevant step in advancing knowledge in this field. Therefore, the manuscript would benefit by including more reflections and literature about the fact that academia suffers from the myth of being a gendered organization (Acker, J. 1990. Hierarchies, jobs, bodies: A Theory of Gendered Organizations; Hearn, J., Husu, L. 2011.

Understanding gender: Some implications for science and technology; Husu, Liisa (2013). Interrogating gender paradoxes in changing academic and scientific organizations.

And other literature targeting these issues in academia (Leathwood, C., Read, B.

2008. Gender and the Changing Face of higher Education. A Feminized Future?; Husu, L. 2005Women's Work-Related and Family-Related Discrimination and Support in Academia; Carvalho & Diogo (2018). Women rectors and leadership narratives: The same male norm?

And there is also more literature that problematizes the argument that women tend to spend more time on teaching and less on research, cf. e.g., Carvalho & Diogo (2021) Time and Academic Multitasking–Unbounded Relation Between Professional and Personal Time. these are just some references that can help to ground the theoretical contributions of this work.

The english can be improved, although it is perfectly understandable.

Other minor things: Page 8 abstract

Lines 27 and 28: “After transcription, the data was thematically analysed using an inductive process. The Nvivo 12 software was used”. This might be left outside the abstract and be included only in the methodology section

Page 9, line 47 – include the acronym (STEM) at the end of that statement (before reference [2]).

These are just small details

Reviewer #2: The study addresses a very timely topic, that of gender inequalities in Higher Education (HE) and Research and Innovation (R&I) institutions which has been on top of discussion in recent years both in academic research and in public debate. The paper tackles two key areas - gender equality in careers and in decision-making bodies - in five West African countries (Ghana, Senegal, Burkina Faso, Niger and Mali). Specifically, it has two main goals: to identify obstacles to the career advancement of women in HE and R&I institutions and to identify strategies to achieve a more equitable representation of women in those institutions. To address these research goals, thirty in depth interviews have been conducted in research institutions of these five West African countries.

Overall, the language is clear, the paper is well organized and makes an important contribution to the literature on gender equality in the African context which is considerably less developed than that focusing on European or North-American cases.

Nonetheless, while the study appears to be sound, it fails to relate to previous research in this area both regarding gender equality in the specific institutional context of the academia, and also concerning gender equality in the context of West African societies. Therefore, I suggest that both the Introduction and the Discussion are reviewed and possibly rewritten to take previous research into account.

Next, I make some detailed suggestions regarding each of the main aspects of the study:

1 - Research question and theoretical framework:

The study would benefit from a narrower research question focusing primarily on the barriers or obstacles to a more equitable representation of women and men in HE and R&I institutions. This should then be analyzed considering the vast literature on multidimensional explanations for gender inequalities, namely the role of factors at the institutional, cultural and individual levels. More specifically, relevant research on gender equality in HE and R&I institutions is currently under development (see, for example, international projects: “ACT”, “UNiGUAL”, “CHANGE”, among others) and have already produced important outputs that can be incorporated in the study's theoretical and analytical framework. In fact, in the Results sub-sections (pages 8 to 15) different types of obstacles are rightly identified: at the organizational, family and individual levels. These are indeed part of multidimensional analyses of gender inequalities and should be connected to the literature reviewed in the Introduction so that there is a dialogue between previous research and the results of this particular study.

2 - Case selection and justification:

As most research on the topic is based on cases from the global north, it is crucial that the study provides an adequate background for the cases it addresses, namely the five selected West African countries. It would be useful to point out not only the gender patterns in academia (which is briefly done only in the cases of Mali and Senegal) but also more general gender equality indicators as provided by different international organizations. This contextualization of the selected cases would ideally include information on policy measures and actions adopted by HE and R&I institutions regarding gender equality in these countries. In this sense, I suggest the inclusion of some references to secondary literature on these topics in the selected country cases. This initial framework would then allow for the definition of the study’s main expectations regarding the barriers to gender equality in research careers and in decision making bodies which are considerably underdeveloped.

3 - Methods:

The methodological section is clear, organized and technically sound, although I do not consider absolutelly necessary the subdivion in "study design", "study population", "sampling" and "data collection and analysis", as in pages 4 to 6. Additionally, some minor issues can be improved as in the case of the first paragraph of the Results section (page 6) which should be part of the sample description, on page 5.

4 - Results and Conclusions:

The results of the content analysis of the interviews are clearly presented and support the study’s conclusions. However, as previously mentioned, it would benefit from a clearer and more rigorous dialogue with the Introduction. As an example, on page 15, the three initial sentences of the Discussion report to three key aspects underpinning the investigation: the research problem, the theoretical framework and the selected cases. However, neither of these aspects had been sufficiently covered beforehand and the reader feels the need to go back to the introductory section seeking for more information on these points which are only mentioned on a very superficial manner.

6. PLOS authors have the option to publish the peer review history of their article (what does this mean?). If published, this will include your full peer review and any attached files.

Reviewer #1: No

Reviewer #2: No

---

## [Author Response · Author response to Decision Letter 0]

18 Oct 2021

Responses to Reviewers 

Reviewers 1 : 

The discussion was reinforced from a theoretical perspective. Fagenson's theory was used as a theory framework. The literature suggested by reviewer 1 has been exploited and integrated as being relevant to the work done in our introduction.

Lines 27 and 28: the sentence referring to a part of the methodology in the abstract has been removed. It is found in the methodology

Page 9, line 47: the acronym STEM has been introduced before reference 2 

Reviewer 2

1- The context was enhanced with data on African academies and their gender policies. Previous research has been integrated into the study. 

2- Previous studies concerning the themes have been included in the study's theorical and analitycal framework. The introduction has been improved by the latter allowing a better understanding of the research results. 

3- A better contextualization of African research institutes was done. Policies regarding gender equality in African research spaces, African institutes and the countries concerned by the study were made. 

4- The methodology section has been improved in terms of form. The subdivisions (study design, study population, sampling and data collection and analysis) and have been removed 

5- The introduction has been improved to better support the results and conclusions of the study

Editor’s comments 

1. We have respected PLOS ONE’s style requirements 

2. The participation of researchers from different West African countries in this research is governed by the provisions of the West African Network of National Ethics Committees of Research for Health (WANEC), of which the National Ethics Committee of Research for Health of Senegal is a member

3. We include more information on the number of interviewers and their characteristics

4. Reference list are reviewed

---

## [Decision Letter · Decision Letter 1]

2 Mar 2022

Barriers of West African women scientists in their research and academic careers: A qualitative research

PONE-D-21-20760R1

Dear Dr. ,

We’re pleased to inform you that your manuscript has been judged scientifically suitable for publication and will be formally accepted for publication once it meets all outstanding technical requirements.

Kind regards,

Juliet Kiguli, MA, PhD

Academic Editor

PLOS ONE

Additional Editor Comments (optional):

Waiting for further decision from the Editorial office. Thank you for the manuscript.

Reviewers' comments:

Reviewer's Responses to Questions

**Comments to the Author**

1. If the authors have adequately addressed your comments raised in a previous round of review and you feel that this manuscript is now acceptable for publication, you may indicate that here to bypass the “Comments to the Author” section, enter your conflict of interest statement in the “Confidential to Editor” section, and submit your "Accept" recommendation.

Reviewer #1: All comments have been addressed

Reviewer #2: All comments have been addressed

2. Is the manuscript technically sound, and do the data support the conclusions?

Reviewer #1: Yes

Reviewer #2: Yes

3. Has the statistical analysis been performed appropriately and rigorously? 

Reviewer #1: Yes

Reviewer #2: N/A

4. Have the authors made all data underlying the findings in their manuscript fully available?

Reviewer #1: Yes

Reviewer #2: Yes

5. Is the manuscript presented in an intelligible fashion and written in standard English?

Reviewer #1: Yes

Reviewer #2: Yes

6. Review Comments to the Author

Reviewer #1: (No Response)

Reviewer #2: (No Response)

7. PLOS authors have the option to publish the peer review history of their article (what does this mean?). If published, this will include your full peer review and any attached files.

Reviewer #1: No

Reviewer #2: No

---

## [Editor Report · Acceptance letter]

4 Mar 2022

PONE-D-21-20760R1 

Barriers of West African women scientists in their research and academic careers: A qualitative research 

Dear Dr. Sougou:

I'm pleased to inform you that your manuscript has been deemed suitable for publication in PLOS ONE. Congratulations! Your manuscript is now with our production department. 

Kind regards, 

on behalf of

Dr. Juliet Kiguli 

Academic Editor

PLOS ONE